# Human Parainfluenza Virus (HPIV) Detection in Hospitalized Children with Acute Respiratory Tract Infection in the Western Cape, South Africa during 2014–2022 Reveals a Shift in Dominance of HPIV 3 and 4 Infections

**DOI:** 10.3390/diagnostics13152576

**Published:** 2023-08-02

**Authors:** Jane Parsons, Stephen Korsman, Heidi Smuts, Nei-Yuan Hsiao, Ziyaad Valley-Omar, Tathym Gelderbloem, Diana Hardie

**Affiliations:** 1Division of Medical Virology, Department of Pathology, University of Cape Town, Cape Town 7700, South Africa; jane.parsons@biomerieux.com (J.P.); stephen.korsman@nhls.ac.za (S.K.); heidi.smuts@uct.ac.za (H.S.); marvin.hsiao@uct.ac.za (N.-Y.H.); z.valley-omar@uct.ac.za (Z.V.-O.); 2National Health Laboratory Service, Johannesburg 2193, South Africa; tathym.gelderbloem@nhls.ac.za; 3Wellcome Centre for Infectious Disease Research in Africa, Institute of Infectious Disease and Molecular Medicine, University of Cape Town, Cape Town 7700, South Africa

**Keywords:** human parainfluenza virus, epidemiology, South Africa, children, multiplex real-time PCR, acute respiratory infection, pneumonia

## Abstract

The epidemiology of human parainfluenza viruses (HPIV), particularly its role as a cause of acute respiratory infection (ARI) in infants, has not been formally studied in South Africa. We evaluated HPIV prevalence in diagnostic samples from hospitalized children from public sector hospitals in the Western Cape between 2014 and 2022. HPIV infection was detected in 2–10% of patients, with the majority of infections detected in children less than 1 year of age. Prior to 2020, HPIV 4 (40%) and HPIV 3 (34%) were the most prevalent types, with seasonal peaks in late winter/spring for HPIV 3 and autumn/winter for HPIV 4. HPIV 4A and 4B co-circulated during the seasonal activity between 2014 and 2017. Pandemic restrictions in 2020 had a profound effect on HPIV circulation and the rebound was dominated by waves of HPIV 3, accounting for 66% of detections and a sustained decline in the circulation of HPIV 1, 2 and 4. An immunity gap could account for the surge in HPIV 3 infections, but the decline in prior HPIV 4 dominance is unexplained and requires further study.

## 1. Introduction

Lower respiratory tract infections (LRTIs) are the leading cause of death globally in children under 5 years [1]. The PERCH study, using (amongst other investigations) a real-time multiplex PCR assay that detected 33 viral, bacterial and fungal targets, showed that in low- and middle-income countries (LMICs) viruses are the predominant cause of LRTIs, requiring hospitalization of young children and that a small number of pathogens is responsible for most cases [2]. While respiratory syncytial virus (RSV) is recognized as the leading cause of pneumonia in young children, the PERCH study showed that 10 organisms account for 79–90% of pneumonia cases and the presence of co-pathogens is associated with increased probability of disease [2]. Among the other common respiratory pathogens, human parainfluenza viruses (HPIVs) as a group were implicated as the third or fourth leading cause of pneumonia in most study sites in the PERCH study [2,3], but has received much less attention from both clinicians and researchers, particularly in the LMIC setting.

Four serotypes of human parainfluenza viruses, HPIV 1–4, all belonging to the *Paramyxoviridae* family, were originally described taxonomically. These four HPIVs are now recognized as different viruses genotypically, with HPIV 1 and 3 belonging to the genus *Respirovirus* and HPIV 2 and 4 belonging to the genus *Orthorubulavirus* [4]. HPIV 4 has two further antigenically distinct subtypes, namely A and B, but these are closely related and do not constitute separate species. 

The global burden of HPIV associated with acute LTRI is widely under-recognized. In a systematic review, Wang et al. [5] estimated the global burden of HPIV infection in children to be approximately 13% of all acute LRTI cases, 4–14% of hospital admissions for LRTI, and 4% of childhood mortality from acute LRTIs. 

With rising clinical use of multiplex PCR assays in LRTI, analysis of diagnostic data is providing new insights into the seasonal prevalence of respiratory viral pathogens. In this study, we report on the prevalence of the HPIVs 1–4 in diagnostic samples over an 8-year period (2014–2022). We aimed to improve our understanding of the epidemiology of HPIVs in Cape Town, Western Cape, South Africa (SA), especially that of HPIV 4 which has not been widely studied. We also aimed to describe the molecular epidemiology of HPIV 4 during the pre-pandemic period and determine the impact of the pandemic on HPIV epidemiology in symptomatic children during the lockdowns and immediately after restrictions were lifted.

## 2. Materials and Methods

### 2.1. Study Design

We conducted a retrospective study using routine diagnostic test results and stored samples to describe the epidemiology of respiratory viruses detected in hospitalized children with acute respiratory infections (ARIs) in Cape Town, South Africa. This study was performed in the Groote Schuur Diagnostic Virology laboratory which provides laboratory services to public healthcare sector patients in the Western Cape province of South Africa. Only samples referred to and tested at this laboratory were analyzed in this study. The results of diagnostic respiratory multiplex PCR were retrieved from the laboratory information system. Any data from individuals aged 18 years and above were excluded from the study.

### 2.2. Testing of Clinical Samples

Sample types typically received for respiratory virus testing included sputum, induced sputum, nasopharyngeal aspirates, or nasopharyngeal swabs from live patients and tracheal and lung swabs or lung tissue on post-mortem cases. Sample processing depends on sample type, swabs are added to viral transport medium or saline, while mucoid samples are diluted, heated and liquified prior to nucleic acid extraction. Nucleic acid extraction was performed using the Easymag system (BioMerieux, Craponne, France). Multiplex PCR amplification was performed on a BioRad CFX thermal cycler, according to the manufacturer’s protocol. Between February 2014 and March 2020, the Anyplex RV16 multiplex PCR assay (Seegene, Seoul, Republic of Korea), which tests for 16 common respiratory viruses (influenza A and B, RSV A and B, parainfluenza viruses 1–4, human metapneumovirus, adenovirus, human bocavirus, enterovirus, rhinovirus, human coronaviruses 229E, NL63 and OC43), was used. This assay specifically identifies the four distinct HPIVs. Samples received after March 2020 were tested with the Allplex Essential respiratory virus assay (Seegene, Seoul, Republic of Korea) which tests for influenza A, influenza B, RSV (A and B undifferentiated), parainfluenza virus (1–4 undifferentiated), human metapneumovirus, rhinovirus and adenovirus. The change to the Essential assay for screening patients was prompted by the better turn-around times achievable with the Essential assay, with minimal loss of clinically important virus targets. Both assays have been extensively validated and yield concordant results with each other and other equivalent assays [6,7].

### 2.3. Typing of HPIV Positive Samples Collected during 2020–2022

HPIV positive samples collected between March 2020 and December 2022 required further testing to identify the specific virus present in the samples. Stored extracted nucleic acid was retrieved and was tested with either the Anyplex RV16 assay or Allplex respiratory panel 2 (Seegene, Seoul, Republic of Korea) assay to type the virus in the sample. Samples typed were selected on the basis of availability and those with CT <38 on RV Essential assay. Altogether, 25 of 29 HPIVs from 2020, 68 of 94 from 2021, and 154 of 175 for 2022 were typed. 

### 2.4. Subtyping of HPIV 4 Positive Samples Collected during 2014–2017

In addition, subtyping was performed on 99 of 282 (35.1%) HPIV 4 positive samples collected between 2014 and 2017 using an in-house developed assay and partial gene sequencing of the HPIV 4 HN gene [8]. The HN protein is the major site of antigenic variation and the gene of choice for phylogenetic studies [9,10,11]. 

### 2.5. Phylogenetic Analysis of HPIV 4 Positive Sequences

A hemi-nested PCR protocol was used to amplify a 777-base pair (bp) region of the hemagglutinin (HN) gene, targeting the globular head of the viral glycoprotein. (Refer to Appendix A for primer sequences). Sanger sequencing of PCR products was performed, followed by alignment with reference sequences from GenBank using ClustalW Multiple alignment with the BioEdit version 7.2.5 [12]. Sequences of clinical samples were de-identified and assigned codes. Using MEGA6, an alignment of 713 nucleotides was analyzed with the neighbor joining algorithm [13]. Robustness was tested via boot strapping 1000 replicates and values of 80% or greater resulted on the branch nodes of phylogenetic trees.

### 2.6. Comparing Respiratory Virus Data Generated from Anyplex RV16 and Allplex Essential Assays

To normalize for the change in assay usage (RV16 before March 2020 and Allplex Essential assay after March 2020), only the viruses detected via both assays are compared in this analysis. Comparison of the proportions of viruses detected in the 2 time periods, 2014–2019 and 2020–2022, was conducted using the total number of viruses detected during each period as the denominator. 

## 3. Results

### 3.1. Respiratory Virus Detections during 2014–2022

In total, real-time PCR results of 18,822 respiratory virus were retrieved from the laboratory information system. From 2014 to the end of 2019, 7316 of 11,584 (63.2%) samples had tested positive for at least one respiratory virus, with results normalized to the same viruses as detected using the RV Essential kit. Human rhinovirus (37%), adenovirus (26%), RSV (19%) and HPIV (9%) were the four most commonly detected respiratory viruses (Figure 1a). From March 2020 to the end of 2022, 3013 of 5694 (52.9%) samples were positive when tested using the RV Essential kit, with the lowest detection rate of 43.8% recorded in 2020 (when COVID restrictions were most stringent) (Appendix A). While there was a distinct decrease in virus detection rates in the period from 2020 to 2022, overall, the proportions of different viruses detected did not change (Figure 1b). Greater granularity on virus detection rates by month and year will be expanded on later in Figure 5) but it can be noted that from 2020 to 2022, there were substantial changes in monthly detection rates of most viruses.

### 3.2. HPIV-Positive Samples

Overall, HPIVs accounted for 9.0% of positive detections. The proportion of samples positive for HPIV varied from year to year, from a low percentage of 2.1% in 2020 to a high percentage of 10.1% in 2014 (Figure 1c). 

The median age of HPIV-positive patients remained similar throughout the period of monitoring, at 0.90 years (IQR 0.46–1.95) before 2020 and 0.82 (IQR 0.41–1.53) between 2020 and 2022 (Appendix A). The median age of infection with RSV was consistently younger, 0.36 years (IQR 0.14–0.96) before COVID and 0.48 (IQR 0.17–1.40) between 2020 and 2022, while the median age of patients infected with adenovirus was older, 1.08 (IQR 0.57–2.05) before 2020 and 1.28 (IQR 0.73–2.29) between 2020 and 2022 (Figure 1d and Appendix A).

### 3.3. HPIV Co-Infections

Co-infecting viruses were frequently present in HPIV-positive samples. Overall, in 42% of samples, HPIV was the only virus detected; in 36%, HPIV plus one additional virus was detected and in 22% of samples, two or more additional viruses were detected. The most frequent co-pathogen was rhinovirus (42%), followed by adenovirus (35%), and RSV (16%). However, they were in proportion to the overall prevalence of these infections, so they do not reflect a significant association (Appendix A).

### 3.4. HPIV Activity during 2014–2019

#### 3.4.1. Type Specific Prevalence

Overall, between 2014 and 2019, HPIV 4 (40%) was most frequently detected in clinical samples, followed by HPIV 3 (34%), and HPIV 1 and 2 both at 13% (Figure 2a,c). When analysed according to age cohort, HPIV 3 at 47% was the most prevalent type detected in the youngest (0–6 month) age group, dropping to 38% in 6–12-month-olds, 27% in 1–2-year-olds, 21% in 2–5-year-olds and 20% in children aged between 5 and 18 years. In contrast, the frequency of HPIV 2 increased with age, from 8% in 0–6-month-olds to 29% in 5–18-year-olds. The proportion of infections caused by HPIV 4 also increased with age in young children (up to the age of 5 years). 

Both HPIV 3 and 4 viruses showed distinct seasonality, but with different seasonal peaks: HPIV 3 was most prevalent in samples collected in late winter and spring (July to November) and HPIV 4 in late summer to early winter (February to July). HPIV 1 and 2 had no obvious seasonal spikes in activity during these years, but rather had low level presence throughout the year. All four viruses had low activity during the mid-summer period (December and January). The years 2016 and 2019 were periods of low activity for all four HPIV viruses (Figure 3a).

#### 3.4.2. HPIV 4A and B

HPIV 4 subtyping data which were available from 2014 to 2017 showed that HPIV 4A and 4B viruses co-circulated during the seasonal peaks of HPIV 4 activity. HPIV 4A (70 of 99 viruses subtyped) was the most prevalent subtype during this period, especially in 2014 and 2017, while HPIV 4B (29 of 99) was present at similar frequency in all four years evaluated (Figure 3b).

#### 3.4.3. Phylogenetic Data on HPIV 4A and B (2014–2017)

Of the 75 HN sequences obtained from the clinical samples that were of adequate quality and length to be included in the alignment, 49 were HPIV 4A and 26 were HPIV 4B. 

HPIV 4A sequences formed two clusters, with other surrounding study sequences which did not form a definite cluster (Figure 4a). Cluster 1 contained samples mainly from 2015 and a single sample from 2016, while cluster 2 samples were mainly from 2017. The un-clustered samples included most of the samples from 2014, but also contained samples from all the other years. The HPIV 4B tree (Figure 4b) shows all study sequences falling into two separate clusters, of which one (cluster 1) contained all of the 2014 samples; otherwise, both clusters contained sequences from samples collected over all the years evaluated.

Based on the observation of sequences from different years and seasons clustered in this way, we concluded that two or more 4A variants and two 4B variants co-circulated during multiple seasons in this period (Figure 4a,b).

### 3.5. HPIV Activity during 2020–2022 (Pandemic Period)

In 2020, non-pharmacological interventions (NPIs) that were instituted in March to curb transmission of SARS-CoV-2 had a significant effect on interrupting transmission of common respiratory viruses, especially the HPIVs, human metapneumovirus and influenza A and B. Figure 5 compares the percentage positivity of HPIV along with that of other respiratory viruses detected during this period from this site. No samples were positive for HPIV between May and October of 2020. In contrast, rhinovirus, adenovirus and RSV continued to be detected, though the frequency of detections were lower than usual, and all viruses displayed out-of-season spikes in activity after restrictions were lifted (Figure 5 and Appendix A). Phased relaxation of NPIs resulted in the return of HPIV circulation after October 2020, but the customary spring wave in 2020 started late with the percentage HPIV positivity remaining high through the summer and in the first half of winter in 2021 (Figure 5 and Appendix A).

#### Type Specific Prevalence 2020–2022

Altogether, 298 samples in patients less than 18 years old were positive for HPIVs during this time. There was a distinct change in the relative prevalence of HPIV types, with HPIV 3 now being the predominant type with 66%, followed by HPIV 4 with 22% and HPIV 1 and 2 with 2% and 10% (note that proportions are based on the 251 (84%) samples that were typed, as primary testing after March 2020 was conducted on the Essential panel which does not return the HPIV type) (Figure 2b).

When analyzed according to the age cohort (Figure 2d), HPIV 3 was overwhelmingly the most frequent type detected in young children, with 73% in 0–6-month-olds, 62% in 6–12-month-olds, 68% in 1–2-year-olds, 50% in 2–5-year-olds, and 31% in 5–18-year-olds. HPIV 2 showed a rising trend, from 6% in 0–6-month-olds to 38% in 5–18-year-olds. HPIV 4 was less prevalent in all age cohorts than it had been pre-pandemic, but showed no clear trends. 

The first HPIV wave post relaxation of NPI for COVID-19 was caused exclusively by HPIV 3 and lasted from November 2020 to June 2021. A second shorter wave driven by HPIV 4 occurred from March to July 2021 (this reflects the typical seasonal pattern for HPIV 4 in SA pre-2020). After this, HPIVs 1, 2 and 4 were detected at a low frequency from August 2021 to December 2022. A second extended HPIV 3 (outside of normal season) wave occurred from February to December 2022 (Figure 6). 

## 4. Discussion

Historically, diagnosis of viral respiratory tract infection was performed clinically, or using virus culture, which covered a limited spectrum of organisms, to identify the potential pathogens. The growing use of multiplex PCR for routine diagnostic purposes presents an opportunity to use routine data in order to understand the epidemiology of respiratory viral infection across the globe.

HPIV is a recognized cause of both upper respiratory tract (URTI) and LRTI in infants. In the PERCH study, HPIV was the fourth most common pathogen detected at most study sites (including SA) [2,3] and our data from hospitalized infants in the Western Cape in South Africa support this finding. Between 2 and 10% of samples were positive for HPIV during this 8-year retrospective evaluation and in 78% of cases, HPIV was either the only virus detected in the clinical sample, or there was a single co-pathogen (usually rhinovirus or adenovirus). Overall, in approximately half of all cases, infection occurred in patients less than 1 year of age and 25% of patients were less than 6 months of age. These findings were similar in both pre- and post-pandemic years. This prevalence of infection is comparable with findings in other parts of the world where reported prevalence of HPIV in hospitalized children with ARI is estimated to vary between 4 and 14% [5]. The median age of infection of 0.86 years in positive children between 2016 and 2022 is similar to that reported in other low and middle income countries [5,14], but infections in age groups younger than that is seen in more developed nations, where the majority of infections are reported to occur in the 1–5-year age group [14,15,16,17,18]. This could reflect higher transmissibility of infection due to crowded living conditions and greater difficulty in applying measures to limit the spread of infections in lower socio-economic environments. HPIV transmission occurs largely via respiratory droplets amongst individuals in close contact and via contaminated surfaces. There is minimal aerosol transmission [19].

All four HPIVs were detected in the clinical samples. HPIV 3 and 4 were the most prevalent, accounting for 34% and 40% of HPIV-positive samples before 2020, and 66% and 22% after relaxation of NPIs between 2020 and 2022, respectively. Despite its lower prevalence, overall, in the pre-pandemic period, HPIV 3 was the most prevalent, causing infection in very young children throughout the study period, accounting for 47% of infections in the 0–6-month age group (2014–2019) and 73% in this age group between 2020 and 2022. This is concordant with findings in other studies, including one from South Africa, which also suggest that HPIV 3 infections occur early in life [17,20]. HPIV 3 and 4 viruses each displayed distinct seasonality, with HPIV 3 activity mainly in late winter/spring and HPIV 4 mainly in autumn in the pre-COVID period. HPIV 1 and 2 were sporadically present throughout the year and there was low prevalence of all the four viruses in the height of summer (December and January). The seasonality is similar to that described in other parts of the world [14,15,17,18]. However, between 2014 and 2019 the prevalence of HPIV 4 was much higher than has been described in other studies, where it has typically been reported to account for between 12 and 18% of HPIV infections in other parts of the world [18,21,22]. Curiously, the dominance of HPIV 4 declined substantially following resumption of HPIV circulation in November 2020. HPIV 4 subtyping, which was carried out on a proportion of samples between 2014 and 2017, showed that both subtypes of HPIV 4 co-circulated during seasonal activity, though HPIV 4A was the most prevalent overall, accounting for 70% of samples typed during this time. Both HPIV 4 subtypes had variants that co-circulated across multiple seasons between 2014 and 2017 as determined via phylogenetic analysis. Two molecular epidemiology studies, one from Hong Kong in 2009 [23] and the other from the UK in 2013–2017, also found both HPIV 4A and B subtypes to be present during the periods of seasonal activity [24]. In addition, the UK study showed that distinct lineages of HPIV 4A and 4B co-circulated during the seasons monitored in their study [24]. 

The range of measures instituted in SA in 2020 to curb the transmission of SARS-CoV-2, namely confinement to place of residence, travel restrictions, school and non-essential business closures, bans on mass gatherings, social distancing, mask wearing and use of hand sanitation, had a dramatic effect on the circulation of most respiratory viruses, including the HPIVs. During the level 5 lockdown (weeks 13–18) when restrictions were at their height, circulation of HPIVs ceased and only recommenced in November 2020 after most restrictions had been lifted. The effect of lockdowns on virus circulation is reflected in the fact that the overall sample positivity rate dropped from a mean of 63.7% between 2014 and March 2020 to 43.8% in 2020 (Appendix A). In contrast to HPIV, adenovirus, rhinovirus and RSV had continued to be present in the diagnostic samples throughout the restriction period, although pre-pandemic levels and expected seasonal periods of activity of these viruses were affected and only returned to pre-pandemic levels during 2022 (Figure 5 and Appendix A). This resistance to NPI measures of especially non-enveloped viruses during the COVID-19 pandemic has been well documented [25,26] and also probably reflects the less than optimal control that was achievable during this time in communities using the public sector health (and other) services in South Africa. Lower than normal respiratory virus detection in clinical samples could also reflect the fact that access to healthcare services was more difficult during lockdowns and only very sick patients might have made it to hospital. Of the viruses monitored, HPIV was the first to return to its pre-pandemic levels (Figure 5 and Appendix A). The brisk rise in HPIV activity in November 2020 could, in part, be explained by the relaxation of restrictions which coincided with one of the usual seasonal periods for HPIV activity. HPIV 3 was the first HPIV to return and was responsible for two extended waves from November 2020 to June 2021 and from February to December 2022. Interestingly, a similar phenomenon was seen in Korea after NPIs were lifted. In this case, an out-of-season HPIV 3 wave of infection occurred between July and December in 2021 [27]. The decline in HPIV 4 detections after 2020 was striking, with only a short HPIV 4 wave occurring during its usual period of activity (April to June) in 2021. This altered pattern likely reflects an increase in the susceptible population due to reduced virus circulation during the restrictions. Interestingly, the immunity gap appears to have been most marked for HPIV 3. The NPIs most likely limited exposure in the 2020 birth cohort, leading to a resurgence of infections later. The high rate of HPIV 3 infection we observed in the young children (under 2 years of age) supports this interpretation.

The main limitation of this study is that data collected were derived from passive laboratory surveillance and that unknown changes in testing policy over time or variable testing practice across the region could have affected the results. It also only reflects the epidemiology in children with more severe symptoms (as only hospitalized children were sampled). In addition, only a proportion (84%) of HPIV-positive samples between March 2020 and the end of 2022 were typed. This could have influenced the overall proportion of HPIV types reported for this period. 

## 5. Conclusions

We used laboratory data to describe the seasonal prevalence of HPIV in samples from hospitalized children with ARI over an 8-year period, before, during and after the COVID-19 pandemic. Findings confirm that HPIV is the fourth commonest virus detected in these patients in South Africa, accounting for 9% of positive virus detections, with the majority of infections occurring in children less than one year of age. Interestingly, in the pre-pandemic period, HPIV 4 (at 40%) was the most common circulating HPIV type and both HPIV 4 subtypes co-circulated during periods of seasonal activity during this time. As expected, strict lockdown measures had a profound effect on HPIV circulation, but following the rebound, the overall prevalence of HPIV remained similar (9%) to the pre-pandemic period. Curiously, rebound circulation was characterized by a shift in the circulation of previously dominant HPIV 4 to HPIV 3 with the highest proportion of HPIV 3 infections occurring in children under 2 years of age. Further prospective studies and formal surveillance would improve our understanding of HPIV in LMIC.

## Figures and Tables

**Figure 1 diagnostics-13-02576-f001:**
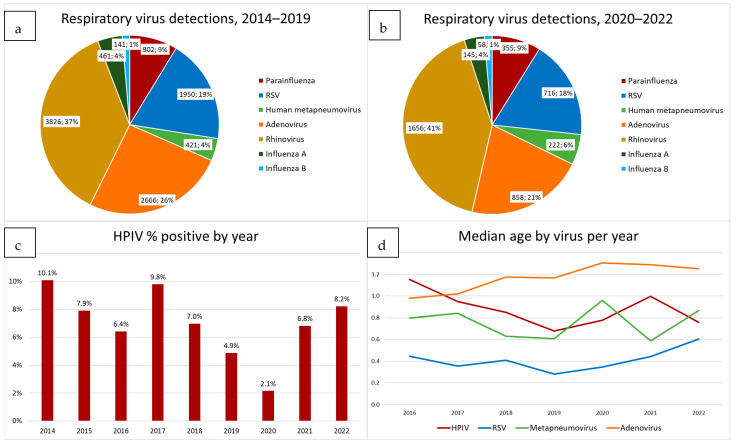
(**a**) shows respiratory virus detections between 2014 and 2019 compared with (**b**) those between 2020 and 2022. Overall, the proportions of detected viruses did not change greatly in the 2 periods. (**c**) shows the percentage positivity of HPIV in clinical samples by year. (**d**) shows median age of children infected by HPIV compared with that for RSV, metapneumovirus and adenovirus. The median age of HPIV infected children is higher than for RSV, but lower than for adenovirus.

**Figure 2 diagnostics-13-02576-f002:**
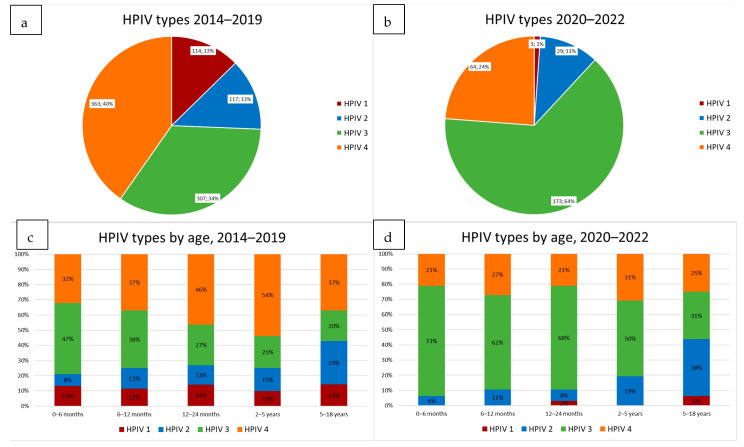
(**a**,**b**) show the prevalence of HPIV types in positive diagnostic samples between 2014–2019 and 2020–2022. A striking change in the prevalence of types 1, 3 and 4 is evident, with a substantial increase in HPIV 3 and a reduction in HPIV 1 and 4 detections in clinical samples during 2020–2022. (**c**,**d**) show the HPIV types detected as a percentage, by patient age for 2014–2019 and for 2020–2022, respectively. HPIV 3 was the most predominant type detected in the 0–6-month age group during both periods, becoming less frequent with increasing age. In contrast, HPIV 2 increased in frequency with children’s age, while HPIV 4 showed no clear trend.

**Figure 3 diagnostics-13-02576-f003:**
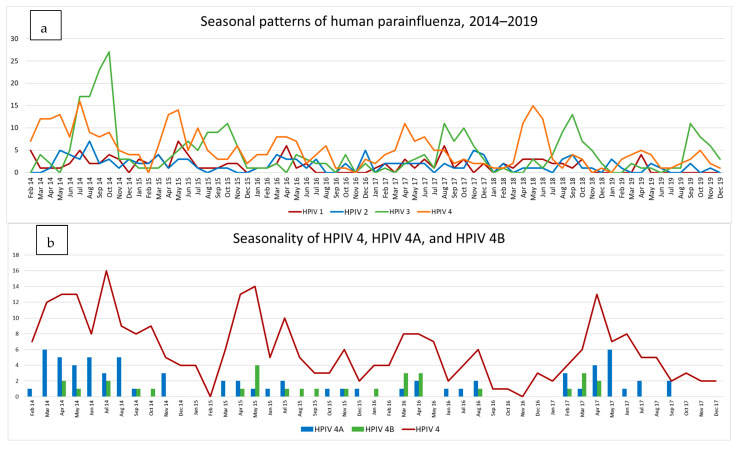
(**a**) shows the seasonal pattern of detection of the 4 HPIVs between 2014 and 2019. The dominance of HPIV 4 and 3 and their distinct seasonality is indicated, with HPIV 3 activity mainly in late winter and spring (July to November) and HPIV 4 in late summer to early winter (February to July). No clear seasonal pattern is evident for HPIVs 1 and 2. (**b**) shows the seasonality of PIV4 between 2014 and 2017 in the line graph. The subtype of samples is indicated by the bars, with blue reflecting HPIV 4A and green reflecting HPIV 4B. HPIV 4A dominates in all 4 seasons. For both, the Y axis indicates number of positives.

**Figure 4 diagnostics-13-02576-f004:**
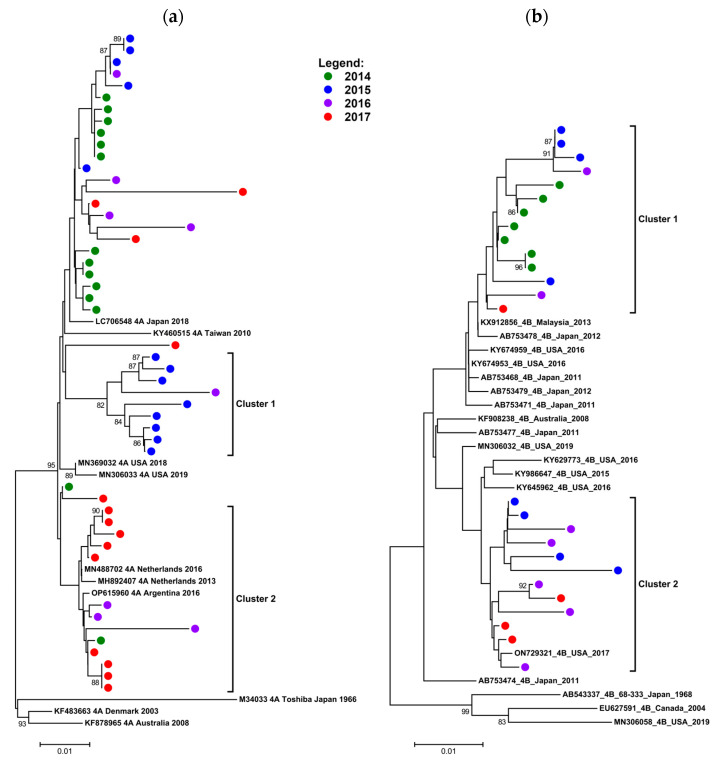
Neighbor joining phylogenetic trees of a 713-base pair (bp) region of the HN gene from HPIV 4-positive diagnostic samples collected between 2014 and 2017 aligned with key reference sequences from GenBank for comparison. The tree in (**a**) shows local HPIV 4A sequences which form 2 distinct clusters with samples from different years, while (**b**) shows local HPIV 4B sequences which also form 2 distinct clusters with sample sequences from different years. Bootstrap support above 80% is shown. Control sequence names show accession number, subtype, country of origin and year of isolate, with the Toshiba (4A) and 68–333 (4B) reference sequences marked with their designation. Study sequences are shown as different colored circles by year as follows: green: 2014; blue: 2015; purple: 2016; red: 2017. Scale bar represents nucleotide substitutions per site. Study sequences have been submitted to GenBank with accession numbers OR176993-OR177061.

**Figure 5 diagnostics-13-02576-f005:**
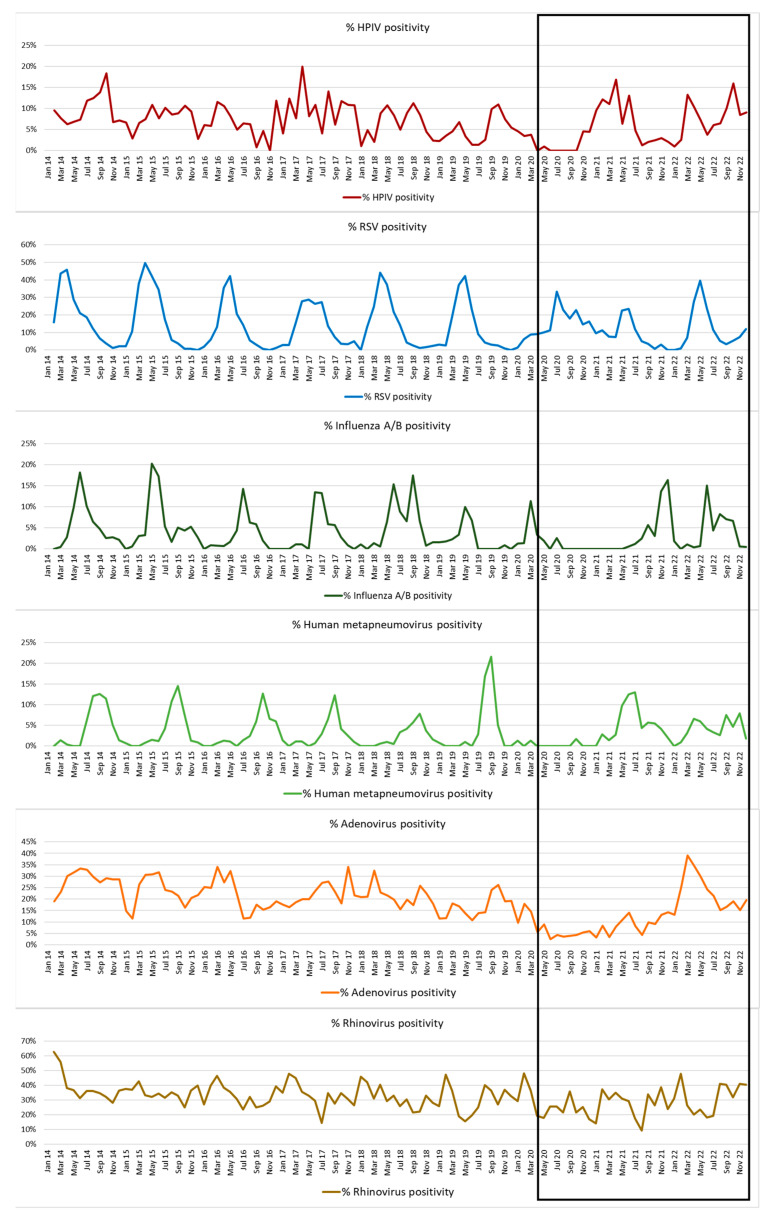
Seasonal epidemiology of HPIV compared with that of other respiratory viruses is shown as percentage positivity of total samples over the 8-year study period: altered seasonality post NPI introduction in March 2020 (boxed section) is noted for all viruses, except possibly for rhinovirus. HPIV rebounds more quickly than influenza, human metapneumovirus, RSV and adenovirus.

**Figure 6 diagnostics-13-02576-f006:**
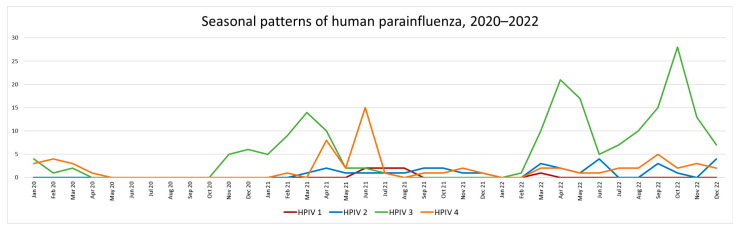
The seasonal prevalence of the 4 HPIVs between 2020 and 2022.shows the dominance of HPIV 3 circulation post lifting of the NPIs. HPIV 4 has a single brief wave from March to July in 2021.

## Data Availability

Sequence data from HN viral gene analysis to be deposited into GenBank, accession numbers OR176993-OR177061.

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
