# Peer review of "Human Parainfluenza Virus (HPIV) Detection in Hospitalized Children with Acute Respiratory Tract Infection in the Western Cape, South Africa during 2014–2022 Reveals a Shift in Dominance of HPIV 3 and 4 Infections"

_diagnostics, 2023, doi:10.3390/diagnostics13152576_

Round 1
Reviewer 1 Report
The authors Jane Parsons et al. in the manuscript entitled “Human parainfluenza virus (HPIV) detection in hospitalized children with acute respiratory tract infection in the Western Cape, South Africa, 2014-2022, reveals a shifting dominance in HPIV 3 and 4 infections evaluated the prevalence in diagnostic specimens collected in recent years of human parainfluenza viruses (HPIV) in the child population and specifically analyzed The 'acute respiratory infection (ARI) in the South African context. The study is well articulated and structured. No minor revision is required.
Author Response
no revisions were requested
Reviewer 2 Report
- “In 2020, non-pharmacological interventions (NPIs) that were instituted in March to curb transmission of SARS-CoV-2 had a significant effect at interrupting transmission of common respiratory viruses, especially the HPIVs, human metapneumovirus and influenza A and B.” What could be the probable reasons for this observation besides social isolation?
2 2. As mentioned, under limitation, were all these samples collected at authorized and affiliated medical laboratories with good storage facilities regarding temperature and maintenance?
3. Did the author examine the family history of Asthma or other respiratory diseases in these subjects?
Author Response
- Probable reasons for the reduction in respiratory virus circulation in 2020: Further details have been added to the discussion (page 11) on the restrictions that were instituted during 2020 to account for how this impacted on the transmission of respiratory viruses.
- Details on the laboratory facilities where samples were processed: all samples were processed only at the one facility, namely Groote Schuur Hospital Diagnostic Laboratory. This has been made clear in the methods section, namely 2.1 on page 2.
- Did the authors examine family history of asthma or other respiratory viruses: We did not have information about the patients, other than age, sex and the fact that they has been admitted to hospital with acute respiratory tract infection. Thus we were unable to assess these question.